

# Behavioral dominance interactions between two species of burying beetles (*Nicrophorus orbicollis* and *Nicrophorus pustulatus*)

Yohanna D. Vangenne, Brendan Sheppard and Paul R. Martin

Department of Biology, Queen's University, Kingston, Ontario, Canada

## ABSTRACT

Closely related species with ecological similarity often aggressively compete for a common, limited resource. This competition is usually asymmetric and results in one species being behaviorally dominant over the other. Trade-offs between traits for behavioral dominance and alternative strategies can result in different methods of resource acquisition between the dominant and subordinate species, with important consequences for resource partitioning and community structure. Body size is a key trait thought to commonly determine behavioral dominance. Priority effects (*i.e.*, which species arrives at the resource first), however, can also determine the outcome of interactions, as can species-specific traits besides size that give an advantage in aggressive contests (*e.g.*, weapons). Here, we test among these three alternative hypotheses of body size, priority effects, and species identity for what determines the outcome of competitive interactions among two species of burying beetles, *Nicrophorus orbicollis* and *N. pustulatus*. Both overlap in habitat and seasonality and exhibit aggressive competition over a shared breeding resource of small vertebrate carrion. In trials, we simulated what would happen upon the beetles' discovery of a carcass in nature by placing a carcass and one beetle of each species in a container and observing interactions over 13 h trials (*n* = 17 trials). We recorded and categorized interactions between beetles and the duration each individual spent in contact with the carcass (the key resource) to determine which hypothesis predicted trial outcomes. Body size was our only significant predictor; the largest species won most aggressive interactions and spent more time in contact with the carcass. Our results offer insight into the ecology and patterns of resource partitioning of *N. orbicollis* and *N. pustulatus*, the latter of which is unique among local *Nicrophorus* for being a canopy specialist. *N. pustulatus* is also unique among all *Nicrophorus* in using snake eggs, in addition to other carrion, as a breeding resource. Our results highlight the importance of body size and related trade-offs in ecology and suggest parallels with other coexisting species and communities.

Corresponding author
Yohanna D. Vangenne,
yohannav@outlook.com

## INTRODUCTION

Competition among closely related and ecologically similar species is common when reproduction and survival depend on limited resources (*Violle et al., 2011*). In animals, the resulting competition is often direct and aggressive; the outcomes are usually asymmetric, with one species being consistently dominant over the other (*Morse, 1974*; *Persson, 1985*; *Martin, Freshwater & Ghalambor, 2017*). Body size is one of the most important traits determining dominance in many species, providing a direct benefit in competitive interactions (*e.g.*, greater inertia, muscle mass), and often correlating with traits useful for fighting (*e.g.*, size of mouthparts, fighting appendages, and limbs) (*Peters, 1983*; *Martin & Ghalambor, 2014*, *2023*). Order of arrival to a resource (priority effects), however, can also determine which species acquires the resource in competitive interactions (*Poulos & McCormick, 2014*; *Fukami, 2015*). The resource is either claimed before any fighting can occur, or the individual that arrives first is better able to fight for and defend its resource (*Fukami, 2015*). Alternatively, traits other than size can provide an advantage in competitive interactions, often being inherent to one species (*e.g.*, superior weapons, greater aggression) and allowing that species to consistently outcompete another. Understanding how different traits (body size, weapons) *vs* priority effects impact competitive outcomes is important for understanding when and why species partition resources in nature (*Martin, 2015*).

*Nicrophorus* burying beetles are an excellent study system for testing among these alternative hypotheses (body size, priority effects, and species identity) to explain the outcomes of interference competition. Seven species co-occur in southeastern Ontario (*Anderson & Peck, 1985*) and aggressively compete for a shared, ephemeral breeding resource—small vertebrate carcasses (*Trumbo, 1994*). Competition is fierce both within and among species, as breeding pairs rarely share carcasses (*Wilson, Knollenberg & Fudge, 1984*; *Trumbo, 1990a*). Coexistence is likely facilitated by different ecological strategies among *Nicrophorus* species that allow them to partition the breeding resource over time (differences in seasonal and diel activity; *Anderson, 1982*; *Anderson & Peck, 1985*; *Trumbo, 1990b*; *Keller, Howard & Hall, 2019*; *Wettlaufer et al., 2021*, *2023*), space (differences in habitat; *Anderson, 1982*; *Anderson & Peck, 1985*; *Burke et al., 2020*), and characteristics of the resource (differences in size of carcass; *Trumbo, 1990a*). The partitioning observed is thought to reflect the different strategies of dominant and subordinate species. This partitioning may also reflect the trade-offs between the traits linked to behavioral dominance associated with each strategy. These trade-offs are also thought to allow dominant and subordinate species to co-occur rather than compete to the point of competitive exclusion (*Grether et al., 2017*; *Martin & Ghalambor, 2023*).

While competitive ability among *Nicrophorus* species varies, the outcomes of aggressive contests seem to be strongly influenced by single traits (*Wilson, Knollenberg & Fudge, 1984*; *Trumbo, 1990a*; *Schrempf et al., 2021*). Previous work shows that larger size is an excellent predictor of behavioral dominance among *Nicrophorus* species in competitive interactions for carcasses (*Wilson, Knollenberg & Fudge, 1984*; *Otronen, 1988*; *Trumbo, 1990a*, *Schrempf et al., 2021*). However, when two individuals of similar size compete, the

beetle from the larger-on-average species often wins (*Otronen, 1988*), suggesting that other traits that differ among species also play a role in competitive outcomes. Alternatively, other evidence suggests that the order of arrival influences the ability to remain in possession of a carcass and defend it (*Otronen, 1988*); however, later arriving dominant species can still usurp a carcass from a subordinate species in some cases (*Wilson, Knollenberg & Fudge, 1984*; *Trumbo, 1990a*). Overall, previous evidence from *Nicrophorus* supports a role for each of our three alternative hypotheses influencing the outcome of competition for carcasses.

In this study, we test among our three alternative hypotheses using two species of *Nicrophorus* burying beetles that are common and co-occur in southeastern Ontario, Canada: *N. orbicollis* and *N. pustulatus*. *N. orbicollis* is on average the largest and most abundant species at our site (*Collard et al., 2021*; *Wettlaufer et al., 2021*). Both species similarly prepare the carcass upon discovery (*Scott, 1998*) and overlap in breeding season (*Anderson & Peck, 1985*; *Wettlaufer et al., 2021*) and habitat. *N. orbicollis* is a habitat generalist (*Burke et al., 2020*), whereas *N. pustulatus* is likely a canopy specialist (*Ulyshen & Hanula, 2007*; *Wettlaufer et al., 2018*). Despite documented use of snake eggs (*Trumbo, 2009*; *Smith et al., 2007*) and even burying of live vertebrates (*DeMarco & Martin, 2020*) for use as breeding resources, *N. pustulatus* uses vertebrate carrion in captivity (*Robertson, 1992*; *Rauter & Moore, 2002*; *Smith et al., 2007*) and appears to compete with other *Nicrophorus* species for vertebrate carrion at our sites (*Wettlaufer et al., 2021*). *N. orbicollis* and *N. pustulatus* are commonly caught in the same traps with the same resource as bait, providing direct evidence for their competitive interactions in nature; at our site in southeastern Ontario, 94 of 118 traps (79.7%) that caught *N. pustulatus* also caught *N. orbicollis*, including 14 of the 16 (87.5%) traps that caught *N. pustulatus* on the ground (*Wettlaufer et al., 2021*).

We used same-sex captive trials involving wild-caught beetles (following *Schrempf et al., 2021*) that replicate one scenario of what can happen when individuals from *N. orbicollis* and *N. pustulatus* come across a carcass in nature. We tested our alternative hypotheses of body size, priority effects and species identity for determining the outcome of interactions for a carcass using summaries of competitive interaction outcomes, where priority is defined as the individual who acquires the resource (carcass) first. We then tested whether the individual that won the most interactions (the dominant) in that trial also spent the most time in contact with the carcass (the resource).

## MATERIALS AND METHODS

### Collection and identification of species

We collected *N. orbicollis* and *N. pustulatus* from 2–14 June, 2021 from mixed forest near Glenburnie, Ontario, Canada (44.302408, −76.515571, 125 m elevation). We used wild-caught beetles, rather than captive-bred beetles, because we did not have access to captively bred individuals of both species, and captive breeding can lead to altered behaviors and other traits, including aggression, compared with wild individuals (*Price, 1999*; *Kelley, Magurran & García, 2006*), which could have altered our results. We set live traps (plastic buckets) baited with raw chicken wings or dead mice set on potting soil and

leaf litter in two different locations: *N. orbicollis* traps on the ground and *N. pustulatus* traps suspended 4–7 m in the forest canopy (following *Wettlaufer et al., 2021*). We checked traps every 2–4 days and identified and sexed all captured beetles following, *Anderson & Peck (1985)*. We collected focal beetles and placed them in individual 4 oz clear plastic containers (Reditainer, model no. RTSC100400) containing moist potting soil. To prevent recapture following trials, beetles were marked on their elytra using a black sharpie after each trial and then released 7.5 km away from the initial capture site. Each individual participated in only one trial.

We did not know the individual histories of the beetles used in our trials. For example, some of our beetles could have mated previously or had previous experience fighting for carcasses. If previous experience varied with the traits of interest (*e.g.*, larger beetles were more likely to have experience fighting), then our results could be biased, leading to incorrect conclusions (*e.g.*, size impacts the likelihood of experience, which then improves fighting success, rather than a direct benefit of large size in fighting). We revisit some of these limitations in our discussion. The time between a beetle's capture and their participation in a trial averaged 2.8 days (range 2–5 days); beetles in the same trial were usually caught on the same day and held in captivity for the same number of days before trials. We held beetles for at least 2 days in captivity before trials because we wanted beetles to readjust their behaviors after having been held within the live traps.

## Captive studies

The methods for our competitive trials follow *Schrempf et al. (2021)*. Briefly, we conducted trials in clear plastic containers (31.4 × 20.8 × 11.9 cm) with 1–2 cm of fresh topsoil. For each trial, we placed a mouse (*Mus musculus*) carcass (previously frozen, then thawed for 48 h at ~22 °C; we removed the tail in most trials to facilitate observations of contact with the carcass) in the center of the container on top of the soil. Mice were sourced from the Queen's University Health Sciences breeding facilities, culled using carbon dioxide gas and frozen until used in our study. Mice used in our trials weighed 30.6–46.5 g (mean weight of 38.5 g).

Beetles were selected for trials based on their sex and collection date—either they were collected on the same day (and thus held for the same amount of time; *n* = 16 trials), or, when this was not possible (*n* = 1 trial), they were collected within 1 day of each other. Within these groups, we selected beetles haphazardly for trials, without regard to size. Just before the start of each trial, we placed one pair of beetles in a container (one *N. orbicollis* and one *N. pustulatus*). Beetles were introduced to the container at the same time, providing equal opportunity to reach the carcass first. All trials consisted of same-sex pairs (see Table 1) to control for sex as a potential confound; however, previous studies have found no evidence for sex affecting the outcomes of aggressive contests between *Nicrophorus* species (*Schrempf et al., 2021*; *Wettlaufer et al., 2023*).

Throughout trials, we used a glass pane to cover the container and red bicycle lights (located within the container to avoid reflectance off the glass) to illuminate the trials at
**Table 1 Sex and morphological measurements of beetles used in experiments.**

| Trial | Trial start date | *N. orbicollis* sex | *N. pustulatus* sex | Mass of *N. orbicollis* (g) | Mass of *N. pustulatus* (g) | Elytra length of *N. orbicollis* (mm) | Elytra length of *N. pustulatus* (mm) | Pronotum width of *N. orbicollis* (mm) | Pronotum width of *N. pustulatus* (mm) |
|---|---|---|---|---|---|---|---|---|---|
| 1 | 04-Jun | Male | Male | 0.440 | 0.239 | 11.2 | 9.2 | 6.5 | 5.7 |
| 2 | 07-Jun | Female | Female | 0.301 | 0.435 | 9.9 | 10.2 | 5.4 | 7.3 |
| 3 | 07-Jun | Female | Female | 0.804 | 0.299 | 13.8 | 9.4 | 7.7 | 6.0 |
| 4 | 08-Jun | Male | Male | 0.458 | 0.365 | 11.8 | 9.8 | 6.6 | 6.3 |
| 5 | 08-Jun | Female | Female | 0.556 | 0.600 | 12.1 | 11.6 | 7.1 | 7.7 |
| 6 | 09-Jun | Female | Female | 0.916 | 0.318 | 15.7 | 9.9 | 8.6 | 6.2 |
| 7 | 10-Jun | Male | Male | 0.509 | 0.484 | 12.4 | 11.4 | 7.5 | 7.1 |
| 8 | 10-Jun | Female | Female | 0.379 | 0.458 | 10.9 | 11.3 | 6.4 | 7.0 |
| 9 | 11-Jun | Male | Male | 0.687 | 0.352 | 13.0 | 9.8 | 7.7 | 6.2 |
| 10 | 11-Jun | Male | Male | 0.212 | 0.375 | 8.7 | 10.4 | 4.8 | 6.4 |
| 11 | 12-Jun | Male | Male | 0.427 | 0.378 | 11.0 | 10.0 | 6.7 | 6.8 |
| 12 | 12-Jun | Male | Male | 0.515 | 0.710 | 12.1 | 12.6 | 7.5 | 7.8 |
| 13 | 13-Jun | Male | Male | 0.354 | 0.650 | 10.2 | 12.6 | 6.1 | 8.3 |
| 14 | 13-Jun | Female | Female | 0.763 | 0.503 | 12.8 | 10.9 | 8.1 | 7.2 |
| 15 | 14-Jun | Male | Male | 0.617 | 0.275 | 13.1 | 9.1 | 7.9 | 5.8 |
| 16 | 14-Jun | Female | Female | 0.705 | 0.529 | 13.5 | 10.4 | 7.5 | 7.2 |
| 17 | 16-Jun | Male | Male | 0.529 | 0.455 | 12.4 | 10.6 | 7.1 | 7.1 |

night, thereby minimizing disturbance to the beetles. Beetles were exposed to natural light from nearby windows at controlled temperatures ranging from 21–24 °C. Trials began at 18:00 and ended at 7:00 (13 h total), and any interactions already underway prior to the start of trials were only timed from 18:00. In total, we conducted 18 trials (one was excluded from analysis because it included beetles of different sexes), each recorded using a Cannon Vixia HF R50 Camcorder (Canon Inc., Tokyo, Japan). We reviewed all trials using VLC media player (www.videolan.org; VideoLAN, Paris, France).

## Size measurements

We used mass as our estimate of relative size of beetles in our trials. We consider mass to be a good proxy of body size (following *Peters, 1983*; see also *Schrempf et al., 2021*; *Wettlaufer et al., 2023*) because different species of *Nicrophorus* may have different morphological proportions that correspond to different ecological strategies unrelated to overall size (*Burke, 2017*). After each trial was complete, we returned the beetles to their plastic containers and chilled them in a cooler filled with ice for 30–60 min. Cooling the beetles slowed their activity, facilitating measurement. Once cooled, we measured each beetle's mass (accuracy of 0.001 g) using a GEM20 Smart Weigh jewelry scale (Smart Weigh Packaging Machinery Co., Ltd., Zhongshan City, Guangdong Province, China), where mass (g) = weight of the beetle (g) × (10 g/weight of the 10 g standard (g)).

## Interactions between beetles

We considered beetles to be interacting when they were within 1 cm of each other because previous work (*Schrempf et al., 2021*) suggested that beetles were aware of each other at this distance. Beetles may have been aware of each other at greater distances than 1 cm, but we found it difficult to determine if this was consistently true in our trials. Our 1 cm cutoff may thus underestimate the number of interactions among individuals, but was nonetheless consistently applied across trials. We recorded the duration of each interaction (rounded to the nearest second), and separated interactions between beetles into four categories: (1) Aggressive physical interactions, (2) Avoidance interactions, (3) Neutral interactions, and (4) Other interactions (*Schrempf et al., 2021*).

*(1) Aggressive physical interactions* were defined as instances where we observed direct, aggressive interactions such as grabbing, biting, chasing, or digging after a buried beetle. We separated the outcomes of these interactions into two groups: symmetric outcomes occurred when no beetle won or lost, and asymmetric outcomes occurred when one beetle won and the other lost. Winners were defined as beetles that retained their position after an aggressive interaction; losers were beetles that retreated from the interaction.

*(2) Avoidance interactions* were cases where one beetle adjusted its behavior to avoid or accommodate the other, for example, when a beetle altered its path (after coming within 1 cm of the other beetle) to avoid the other. The loser in this case was the individual who adjusted its position to accommodate the other beetle.

*(3) Neutral interactions* were defined as cases where individuals were within 1 cm of each other and displayed no signs of aggression or avoidance. These included cases of two beetles passing each other with no alteration of path or apparent aggression, or one beetle passing over another buried beetle without digging after and/or attacking it.

*(4) Other interactions* consisted of interactions where we could not observe the details of the interaction but could tell that an interaction had taken place by observing its result. For example, 'Other interactions' included a beetle retreating from underneath the carcass where the other beetle was located, where we could infer that an interaction had occurred but could not tell whether it involved avoidance, chasing, or physical contact.

When categorizing these interactions, we considered encounters that started as one type and then changed into another type as being two interactions. For example, we recorded a neutral interaction that escalated into a physical one as one neutral interaction and one physical interaction, respectively.

## Contact with the carcass

We noted a simple yes/no if the beetle touched the mouse carcass during each minute of the trial. Tails were removed from the majority of trials and contact with the tail in early trials (where the tail was left on) was not counted.

## Priority effects

We defined 'priority' as the first individual to acquire the resource (the mouse carcass), following previous resource-centered definitions of priority effects (*e.g.*, *Poulos & McCormick, 2014*), which differ from community assembly definitions (*e.g.*, which species

colonizes a new habitat or environment; *Fukami, 2015*). Beetles were released into containers at the same time, and typically walked around the outside of the containers before burying into the soil. We recorded which individual was first to make contact with the carcass as our estimate of priority, testing the idea that the first beetle to claim the carcass would be better able to retain possession of the carcass over time. Our definition of priority in our trials does not encompass all ways that priority effects could impact burying beetles in nature. For example, beetles that come into contact with the carcass first could benefit by partially burying the carcass, or quickly attracting a mate and defending the carcass as a pair; our trials did not test for such benefits that could also be important in nature. Nonetheless, our experiments simulate periods of high *Nicrophorus* activity, with beetles arriving at a single carcass in rapid succession. For example, during *Nicrophorus* removal experiments in 2023 at the Queen's University Biological Station near our field site, we recorded rates of arrival at a single mouse carcass of up to 8.4 beetles/hour (*N. orbicollis*) and 6.4 beetles/hour (*N. sayi*); the rapid arrival of beetles leads to widespread fighting over carcasses.

## Behavioral dominance

We defined the behaviorally dominant beetle in each trial as the individual that won the most interactions (sum of the aggressive physical interactions and avoidance interactions), following *Schrempf et al. (2021)*. We then tested whether relative size (body size hypothesis), being the first to reach the carcass (priority effects hypothesis), or species (species identity hypothesis) best predicted the dominant individuals across trials. We also tested whether the dominant individuals in each trial spent more time in contact with the carcass. We also measured the distance (cm) between each beetle and the mouse carcass at the end of each trial for possible use as an indicator of behavioral dominance in future studies, where the dominant species remains in close contact with the carcass (*Wilson, Knollenberg & Fudge, 1984*; *Schrempf et al., 2021*).

## Statistical analyses

We performed all statistical analyses and plotting in R (version 4.1.1; *R Core Team, 2021*), and followed similar approaches to *Schrempf et al. (2021)*.

We tested which of our alternative hypotheses best predicted behavioral dominance using binomial generalized linear models (glm). Our response variable was the proportion of dominance interactions (*i.e.*, aggressive physical interactions, avoidance interactions, and other interactions with an asymmetric outcome; see Table 2) that each species won, combined into one response variable (*i.e.*, cbind (number of interactions won by *N. orbicollis*, number of interactions won by *N. pustulatus*); following *Crawley, 2013*). We then natural log transformed the mass ratio of our two species (ln (mass in *N. orbicollis*/mass in *N. pustulatus*)). Thus, our final models had relative wins by each species as the response variable, and relative mass and first species to the carcass as predictors with their interaction term. We detected overdispersion in our initial models and corrected the standard errors using a quasi-glm model that incorporated a dispersion parameter (*Zuur et al., 2009*). We further checked the fit of our saturated model by plotting

**Table 2 Frequency of interactions among beetles, order of arrival, and time spent in contact with carcass.** We ran our statistical analyses using the winners of all interactions with asymmetric outcomes as the response variable.

| Trial | Trial start date | Aggressive physical interactions* | Avoidance interactions* | Neutral interactions | Other interactions* | Total number of interactions | Initiator of asymmetric interactions^ | First to carcass | Contact with carcass (minutes)^ |
|---|---|---|---|---|---|---|---|---|---|
| 1 | 04-Jun | 18 (17, 0) | 8 (8, 0) | 59 | 2 (1, 0) | 87 | (21, 4) | orbicollis | (521, 30) |
| 2 | 07-Jun | 4 (0, 4) | 4 (0, 4) | 0 | 8 (0, 8) | 16 | (5, 3) | pustulatus | (158, 299) |
| 3 | 07-Jun | 18 (15, 0) | 2 (2, 0) | 11 | 0 | 31 | (16, 3) | orbicollis | (154, 88) |
| 4 | 08-Jun | 14 (14, 0) | 5 (3, 1) | 12 | 0 | 31 | (14, 4) | pustulatus | (307, 138) |
| 5 | 08-Jun | 9 (0, 9) | 1 (0, 1) | 8 | 0 | 18 | (9, 1) | pustulatus | (116, 397) |
| 6 | 09-Jun | 5 (5, 0) | 1 (1, 0) | 10 | 0 | 16 | (6, 0) | orbicollis | (285, 18) |
| 7 | 10-Jun | 6 (0, 6) | 3 (0, 3) | 9 | 1 (1, 0) | 19 | (6, 4) | pustulatus | (1, 123) |
| 8 | 10-Jun | 25 (1, 22) | 7 (2, 5) | 53 | 18 (0, 11) | 103 | (6, 35) | orbicollis | (173, 334) |
| 9 | 11-Jun | 20 (20, 0) | 2 (2, 0) | 8 | 0 | 30 | (20, 3) | orbicollis | (504, 83) |
| 10 | 11-Jun | 5 (1, 4) | 0 (0, 0) | 6 | 0 | 11 | (2, 3) | pustulatus | (286, 315) |
| 11 | 12-Jun | 28 (24, 0) | 10 (5, 3) | 14 | 0 | 52 | (30, 2) | pustulatus | (492, 88) |
| 12 | 12-Jun | 16 (6, 4) | 11 (0, 11) | 22 | 2 (1, 0) | 51 | (7, 15) | pustulatus | (156, 344) |
| 13 | 13-Jun | 16 (1, 15) | 10 (0, 10) | 13 | 5 (0, 5) | 44 | (4, 22) | pustulatus | (89, 454) |
| 14 | 13-Jun | 28 (24, 2) | 3 (3, 0) | 1 | 0 | 32 | (26, 3) | pustulatus | (307, 383) |
| 15 | 14-Jun | 12 (11, 1) | 2 (2, 0) | 18 | 5 (2, 0) | 37 | (13, 0) | orbicollis | (221, 259) |
| 16 | 14-Jun | 2 (0, 2) | 11 (1, 10) | 11 | 3 (0, 0) | 27 | (6, 5) | pustulatus | (157, 604) |
| 17 | 16-Jun | 7 (7, 0) | 0 | 24 | 0 | 31 | (7, 0) | orbicollis | (367, 442) |

Notes:
* Number of asymmetric interactions won by: (*N. orbicollis*, *N. pustulatus*).
^ Values for: *N. orbicollis*, *N. pustulatus*.

predicted *vs* fitted values, theoretical quantiles *vs* standardized Pearson residuals (qqplots), by examining Cook's distance values, and by plotting model residuals accompanied by a Shapiro-Wilk test of normality.

In our models, the relative mass predictor tested the body size hypothesis, the first species to the carcass predictor tested the priority effects hypothesis, and the model intercept tested the species identity hypothesis; intercepts above zero indicated that *N. orbicollis* was more likely to win interactions when the species did not differ in mass, and below zero indicated that *N. pustulatus* was more likely to win when mass was equal. We ran models with all possible combinations of predictor variables using the *dredge* command from the *MuMIn* package (*Bartoń, 2019*). We then compared the relative performance of these models using quasi Akaike information criterion values (QAIC), where the model with the lowest QAIC score was determined to be the best performing model. We checked the fit of our best-performing model using the same methods described for the saturated model.

We tested whether the dominant individual in a trial spent more time in contact with the carcass using a glm with a Gaussian distribution, with the natural log ratio of time in

contact with the carcass, as: ln ((time *N. orbicollis* was in contact with the carcass +1)/(time *N. pustulatus* was in contact with the carcass +1)) as the response variable, and the dominant species in the trial (either *N. orbicollis* or *N. pustulatus*) as the sole predictor. We checked model fit by plotting predicted *vs* fitted values, theoretical quantiles *vs* standardized Pearson residuals (qqplots), by examining Cook's distance values, by plotting model residuals *vs* the predictor accompanied by a Bartlett test of homogeneity of variance, and by plotting model residuals accompanied by a Shapiro-Wilk test of normality.

We tested if the dominant beetle species in each trial was closest to the carcass at the end of the trial using a glm with a Gaussian distribution, with the natural log ratio of distance to the carcass, as: ln ((distance of *N. orbicollis* to the carcass +1)/(distance of *N. pustulatus* to the carcass +1)) as the response variable, and the dominant species in the trial (either *N. orbicollis* or *N. pustulatus*) as the sole predictor. We also tested if beetles interacted more often when competing for larger mouse carcasses using glms, with the number of interactions as the response variable (either all interactions, or just aggressive interactions, in separate models) and mouse mass as the predictor. We checked model fit for these glms using the same methods described for the time in contact with the carcass analysis, described above.

## RESULTS

### Interactions among species

The total number of interactions occurring in each trial varied, however, neutral interactions were the most common (see Table 2). Most aggressive interactions (aggressive physical interactions and avoidance interactions) had asymmetric outcomes, with one beetle emerging as the winner. The duration of interactions varied, with aggressive physical interactions averaging the longest (mean duration of 18.1 s +/− 28.1 SD), avoidance interactions the shortest (mean duration of 4.03 s +/− 5.74 SD), with neutral (mean duration of 12.1 s +/− 38.5 SD) and other interactions (mean duration of 6.66 s +/− 7.69 SD) falling somewhere in between.

Both *N. orbicollis* and *N. pustulatus* initiated interactions with asymmetric outcomes, with *N. orbicollis* emerging as the winner more often (*N. orbicollis* 179 wins: *N. pustulatus* 126 wins; see Table 2; Fig. 1). Throughout trials, *N. orbicollis* initiated more asymmetric interactions than *N. pustulatus* (198:107; see Table 2). In 14 of 17 trials, the winner of the most asymmetric interactions also spent the most time in contact with the carcass (see Table 2, Fig. 2).

Additionally, the beetle initiating a contest won 80.0% (244/305) of cases and was significantly more likely to win the contest (estimate = 1.40 +/− 0.35 SE, $z = 4.0$, $P = 0.00008$). This pattern did not depend on which species initiated the contest (*N. orbicollis*: estimate = 1.45 +/− 0.38 SE, $z = 3.8$, $P = 0.0001$; *N. pustulatus*: estimate = 1.29 +/− 0.44 SE, $z = 2.9$, $P = 0.0036$). In 14/17 trials, the individual that initiated the most interactions was also the trial winner (*i.e.*, the winner of the majority of interactions in a trial).

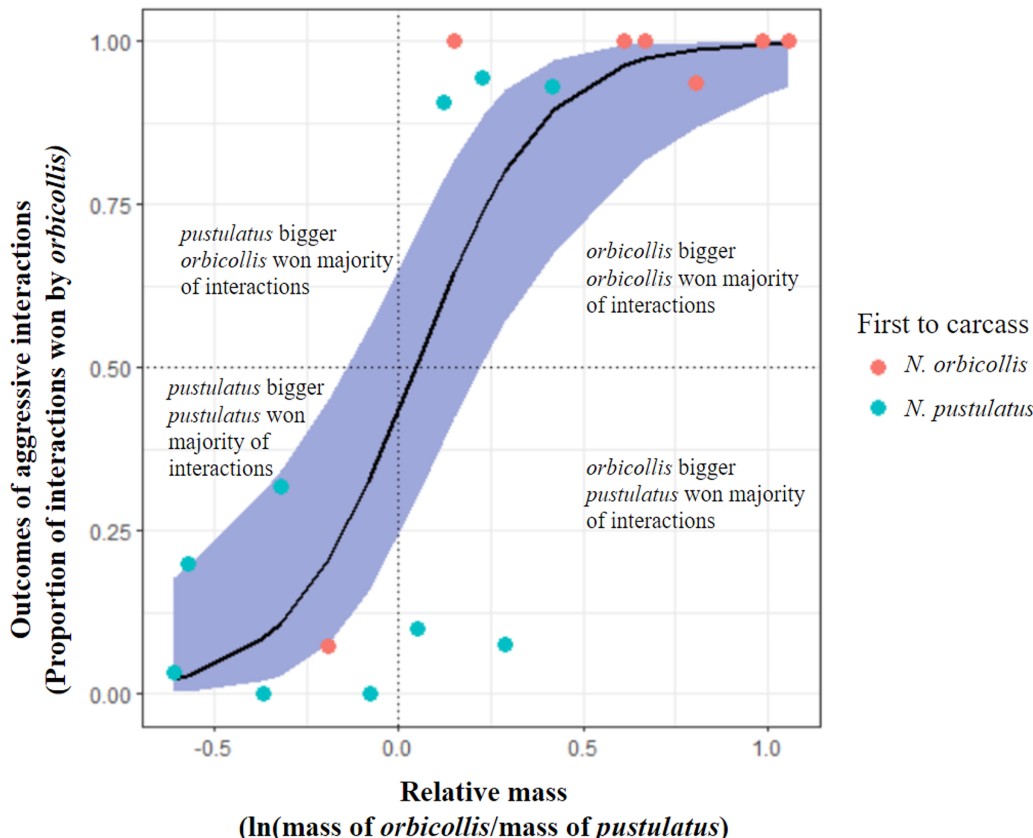

**Figure 1** **Comparison of alternative hypotheses predicting the outcomes of asymmetric interactions.** Graph showing trial outcome (winner as individual who won the most asymmetric interactions; all interactions with asymmetric outcomes included) being significantly predicted by relative mass of individuals. Blue shading represents 95% confidence interval. Points represent trials ($n = 17$) and color of points indicates which species was first to the carcass in each trial (red = *N. orbicollis* first, blue = *N. pustulatus* first).                           

## Factors predicting behavioral dominance

Neither priority effects nor species identity were significant predictors of behavioral dominance between *N. orbicollis* and *N. pustulatus*. Larger relative mass was the only significant predictor of behavioral dominance between *N. orbicollis* and *N. pustulatus* (see Table 3) across interaction types (quasibinomial glm, mass: estimate = 5.76 +/− 1.6 SE, $t = 3.6$, $P = 0.0025$; see Fig. 1). As the difference in relative body size increased, so did the difference in number of wins (see Fig. 1), regardless of species. Restricting the analysis to include only the outcomes of aggressive physical interactions (*e.g.*, fights and chases) resulted in similar results (quasibinomial glm, mass: estimate = 5.85 +/− 1.9 SE, $t = 3.1$, $P = 0.0076$).

## Carcass contact and behavioral dominance

Trial winners predicted which individual spent more time in contact with the carcass (deviance = 22.46, chi-square $P = 0.00005$; see Fig. 2).

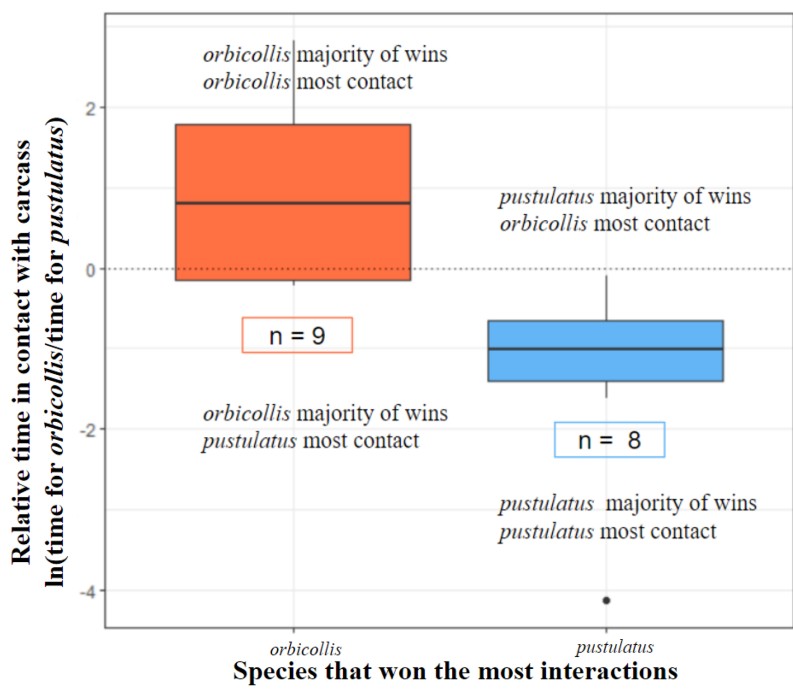

**Figure 2 Species that won the most interactions spent more time in contact with the carcass.** All interactions with asymmetric outcomes included (see Table 2). *N. orbicollis* has nine trial wins and *N. pustulatus* has eight trial wins (*n* = 17). Boxes encompass data from the first quartile to the third. Center lines represent the median. Whiskers above and below represent variability of data outside the upper and lower quartiles. Outlined *n* values indicate how many trials each species won.

**Table 3 Comparison of model performance for quasibinomial generalized linear models.** We evaluated and compared different predictors corresponding to our alternative hypotheses for their ability to predict behavioral dominance among *N. orbicollis* and *N. pustulatus*. The response variable was the proportion of aggressive contests won by *N. orbicollis* or *N. pustulatus* (*n* = 17). Models are ranked by QAICc scores. Values are intercept estimates or effect sizes for predictor variables.

| Model | Intercept | Relative mass | df | logLik | QAIC | ΔQAIC | Weight |
|---|---|---|---|---|---|---|---|
| Outcome~relative.mass | −0.2659 | 5.763 | 2 | −60.205 | 20.6 | 0 | 0.824 |
| Outcome~relative.mass+first.to.carcass | −0.4649 | 5.746 | 3 | −59.926 | 24.1 | 3.43 | 0.149 |
| Outcome~relative.mass*first.to.carcass | −0.4596 | 8.746 | 4 | −56.558 | 27.5 | 6.83 | 0.027 |
| Outcome~1 | 0.2662 | na | 1 | −171.009 | 41.2 | 20.55 | 0 |
| Outcome~first.to.carcass | 0.9416 | na | 2 | −159.139 | 41.7 | 21.02 | 0 |

### Behavioral dominance and carcass proximity

The trial winner was not more likely to be closest to the carcass at the end of a trial ($F = 0.64$, $P = 0.43$; see Table 2).

### Interactions among beetles and carcass size

Larger carcasses did not elicit more interactions among our focal beetles (all interactions, $F = 0.004$, $P = 0.95$; aggressive interactions only, $F = 0.057$, $P = 0.82$).

## DISCUSSION

### Body size predicts behavioral dominance

Relative body mass was our single best predictor of contest outcomes between *N. orbicollis* and *N. pustulatus*, making it our only significant trait determining behavioral dominance. We did not know the histories of wild-caught beetles in our trials (*e.g.*, history of fighting, previous mating); if mass covaried with the history of experience for beetles (*e.g.*, larger beetles had more extensive experience with fighting), then it remains possible that this bias impacted our results. In such a case, the impact of mass would be indirect, manifested through different histories of experience (*e.g.*, fighting experience) or condition (mated or not).

Our findings were largely consistent with previous work examining behavioral dominance interactions within *Nicrophorus*. We confirm body size as the best predictor of behavioral dominance among competing *Nicrophorus* species, including *N. orbicollis* and *N. defodiens* (*Wilson, Knollenberg & Fudge, 1984*), *N. vespiloides, N. investigator*, and *N. vespillo* (*Otronen, 1988*), *N. defodiens, N. orbicollis*, and *N. sayi* (*Trumbo, 1990a*; *Wettlaufer et al., 2023*), as well as *N. orbicollis* and *N. tomentosus* (*Schrempf et al., 2021*). Results are consistent with previous work in demonstrating that neither species identity (*Schrempf et al., 2021*) nor order of arrival affects success in competitive interactions over the carcass among species pairs (*Trumbo, 1990a*; *Wilson, Knollenberg & Fudge, 1984*; *Schrempf et al., 2021*; see *Otronen, 1988* for an exception).

### Benefits of larger size

Aggressive interactions among *Nicrophorus* species typically involve chasing, grappling, digging, and fighting, and thus large size is likely to convey benefits. In all of these cases, greater size could confer greater muscle mass, larger energy reserves, greater inertia, and greater stability when fighting. Large size may also covary with the size and strength of legs and mouth parts that also provide advantages in aggressive interactions by increasing speed, grip, and pushing strength (*Pukowski, 1933*).

Larger relative body size may also signal competitive ability of an individual and play a role prior to the aggressive interactions themselves. The avoidance of larger beetles is supported by the avoidance interactions observed in our trials, where the initiating beetle in avoidance interactions would often turn or adjust its path and move away when approaching within 1 cm of another larger beetle.

The importance of large size in determining the outcome of aggressive contests for carcasses between *N. orbicollis* and *N. pustulatus* is consistent with other previous work on *Nicrophorus* (*Wilson, Knollenberg & Fudge, 1984*; *Otronen, 1988*; *Scott, 1998*; *Trumbo, 1990a*; *Schrempf et al., 2021*; *Wettlaufer et al., 2023*), suggesting a consistent disadvantage to small size in competitive interactions. The advantage of large size in aggressive contests declines over evolutionary time in other groups (birds), where adaptations, such as novel weapons, allow some small species to dominate larger species, and adaptations to specific

challenges, like migration, compromise competitive ability in some larger species (*Martin & Ghalambor, 2014*). In contrast, we find no similar evidence for a decline in the importance of size in aggressive contests among *Nicrophorus*, suggesting that the costs of small size are difficult to overcome. *N. orbicollis* and *N. pustulatus* diverged an estimated 87 million years ago; other species pairs examined to date are also old (>50 million years diverged) (*Sikes & Venables, 2013*), suggesting ample time for the evolution of novel traits that could alter the advantage of large size. Large size, however, may play a more important role in fighting in *Nicrophorus*, or other traits that we did not consider in our study, such as cooperative behaviors (*Trumbo, 1994*), might be key factors providing competitive advantage to some smaller species. Indeed, cooperative behaviors may be more common in smaller species of *Nicrophorus* (*Scott, 1998*).

### Coexistence of *N. orbicollis* and *N. pustulatus*

*N. orbicollis* is larger, on average, than *N. pustulatus* (*Collard et al., 2021*), suggesting that *N. orbicollis* is behaviorally dominant over *N. pustulatus* in most interactions in nature. These results suggest that *N. orbicollis* may preferentially access preferred resources such as the small vertebrate carcasses in the leaf litter of forests, while *N. pustulatus* may be relegated to canopy habitats and alternative resources, such as snake eggs (*Blouin-Demers & Weatherhead, 2000*; *Smith et al., 2007*; *Trumbo, 2009*; *Wettlaufer et al., 2018*; *DeMarco & Martin, 2020*). Such partitioning among closely related species may reflect a trade-off between competitive dominance, mediated by size, and the ability to use canopy habitats and carcasses associated with off-ground vertebrate reproduction (environmental tolerance; *Martin & Ghalambor, 2023*). While our results are consistent with asymmetric dominance interactions creating ecological partitioning among our focal species, we cannot at present rule out alternative hypotheses to explain the pattern, and the factors constraining *N. orbicollis'* success in the canopy remain to be fully explored.

## CONCLUSIONS

Our study reveals that body size is the best predictor of who is behaviorally dominant between individuals of *N. orbicollis* and *N. pustulatus* competing for carcasses, and that priority effects (defined by which species arrives at the carcass first) and species identity are either unimportant or minimally important. Our results support previous work and support body size as a key trait in structuring dominance hierarchies and ecological communities.

## ACKNOWLEDGEMENTS

Thanks to Chris Eckert, the Martin and Bonier labs, and Eryn Basham for their advice and critique. Thanks to Erin Bolger, Katriona Lane, Emma Schubert, David Griffin, Mary Beazely, and James Yates for their help in data collection. We conducted research on traditional Anishinaabe and Haudenosaunee Territory and are grateful to live and learn on these lands.

### Funding
This work was funded by the Natural Sciences and Engineering Research Council of Canada (NSERC) Discovery Grant Program. The funders had no role in study design, data collection and analysis, decision to publish, or preparation of the manuscript.

### Grant Disclosures
The following grant information was disclosed by the authors:
Natural Sciences and Engineering Research Council of Canada (NSERC).

### Competing Interests
The authors declare that they have no competing interests.

### Author Contributions
- Yohanna D. Vangenne performed the experiments, analyzed the data, prepared figures and/or tables, authored or reviewed drafts of the article, and approved the final draft.
- Brendan Sheppard performed the experiments, authored or reviewed drafts of the article, and approved the final draft.
- Paul R. Martin conceived and designed the experiments, performed the experiments, analyzed the data, authored or reviewed drafts of the article, and approved the final draft.

### Data Availability
All raw data is available in the Supplemental Files.

### Supplemental Information
Supplemental information for this article can be found online at http://dx.doi.org/10.7717/peerj.16090#supplemental-information.

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
