# Peer review of "Behavioral dominance interactions between two species of burying beetles (Nicrophorus orbicollis and Nicrophorus pustulatus)"

_PeerJ, doi:10.7717/peerj.16090_

## Round 0.1 · original submission · Major Revisions

Both reviewers were positive about several aspects of your manuscript while also raising some important points that need to be addressed.

First, Reviewer 1 raises an important issue - this manuscript is very similar in many respects to Schrempf et al. 2021 PeerJ 9:e10797. I would think a larger publication studying the interactions of all species would be a stronger paper than publishing what seem to be least-publishable units.

Second, both reviewers question the measurement and use of "priority" in these current study given that members were simultaneously introduced in each trial. Your methodology and interpretation of the observed effect of this variable need to be justified.

Third, Reviewer 1 requires some discussion of the potential biases in using wild-caught individuals in the study.

Finally, Reviewer 2 raises a number of statistical issues that need to be addressed. I look forward to reading a revised version of your manuscript.

Reviewer 1 ·

Basic reporting

The manuscript was well written. The authors used clear hypotheses and a good structure. I have some concerns about the manuscript, however. First, it seems like a carbon copy of a paper from the same group published in this same journal in 2021 (Schrempf et al. 2021 PeerJ 9:e10797), but using a different burying beetle species. The introduction and literature cited seem identical. On this point alone it is not worthy of rejection, however. The science is sound, but one wonders if a paper will result from replicating this study with every species pair that coexist at the authors’ field site.

Second, in the introduction the authors discuss weaponry as a species-level trait that could influence the outcomes of behavioural interactions. While a good example, I think the authors laboured this point—burying beetles do not possess weaponry that is fundamentally different between species. Mandibles are used in fights, but different species do not have drastically different mandibles.

Third, priority is not a trait but is referenced as one throughout the manuscript. In addition, there is nothing in the methods or results about these priority effects. How was it measured or manipulated? I expected the experimental design to include some period of time where an individual was left alone on a carcass to mimic arriving first, with the competitor (smaller or larger) introduced later. Were both individuals introduced at the same time? Any information about priority needs to be rewritten and framed appropriately.

Experimental design

As noted above, the design of the experiment is not clear regarding the priority effect aspect of the manuscript. If the beetles were introduced at the same time, and priority measured by the first to the carcass, I would not interpret this data as any priority effect.

All the beetles used in this manuscript were wild caught individuals. The authors did not have any knowledge of whether the individuals they caught were mated or not, or their previous experience with fighting for resources. Could the authors speculate about how these factors, while unknown, could contribute to the variation in the results they found? With a relatively small sample size, it is possible by chance that their smaller individuals were less likely to have mated which may have been driving the pattern over and above the effect of body size. Some acknowledgement of these issues would be welcome. Why were lab-reared individuals not used?

L150: How did carcass size influence the rate of interactions? Are larger carcasses more fiercely fought over?
L159: Did the individuals have any time in the lab to acclimate?
L170: I do not understand the mass measurement; why is it multiplied by 10g?
L174: Why is 1cm the threshold used for an interaction? Is there any data on the distance when behaviours change in these species? Individuals more than 1cm away might already exhibit different behaviours if they are aware of heterospecifics.

Validity of the findings

This manuscript replicates a previous paper from the same group but has replaced one of the burying beetle species with another. There is little discussion about why the species pair in this manuscript should differ in the mechanism underlying dominance to the previous species pair of N. tomentosus and N. orbicollis, where similarly body size alone was the significant predictor of dominance.

The first three paragraphs of the discussion have considerable repetition about body size and dominance. This information should be one paragraph maximum. The discussion is also lacking in broader conclusions. Are the two species tested here very similar in size, or closely related phylogenetically relative to previously published work? What does this paper add to the understanding of mechanisms of dominance?

All data and code are provided and well annotated.

Additional comments

No comments.

Reviewer 2 ·

Basic reporting

This manuscript is clearly written and provides a nice summary of the basic questions it attempts to answer. The authors provide sufficient background to put the manuscript in a broader context. The paper follows a standard structure and the raw data has been provided (along with a README file). The manuscript I also self contained and for the most part, the conclusions follow from the results.

There is a typo on line 320: "influences" should be "influence".

Experimental design

The study design is generally sound and the methods are fairly easy to follow. I do have a few comments / suggestions for the authors.
1. I suggest dropping trial 152… it is the only one in which the two beetles were not of the same sex. I imagine that this trial involved a mis-sexed beetle (this happens!). In any case, I suggest removing it form the dataset.
2. Line 157- this suggests that the bicycle light was in the container with the beetles and the carcass. Is this correct?
3. It is not clear how the pairs were established for their trials. Were the they assigned randomly or were they chosen to generate variation in the body size ratios. I assume the former, but this is not made clear.
4. It sounds as though the beetles were placed in the container at the same time. Doesn’t this simulate arriving at the carcass simultaneously (or at least nearly so)? In a more natural situation, the beetles might arrive at the carcass hours apart, thus one species may have begun to prepare the carcass before the other arrives. I wonder whether the authors experiment really simulates that type of priority effects that occur in nature.
5. This is kind of a minor point, but the authors refer to their study as an "experiment", however nothing is actually manipulated here. Instead, it is an observational study that is conducted under controlled lab conditions.

Validity of the findings

The analyses and results seem sound. However, I have a few general comments / questions.
1. Lines 248-253: there is a multiple testing issue here (e.g. using three different measures of relative body size, all of which are probably highly correlated, in different models). Have the authors considered adjusting their alpha for this? More generally, I would suggest just picking one measure of size or perhaps constructing a multi-variate measure of size from a PCA.
2. 291-299: A related issue- the authors only present the AIC's for the mass model (Table 3). Is there a reason for this? They also note that for all interaction types the results were the same using different measures of body size. They then restrict the analysis to only aggressive interactions. However, when they do this, they only show the mass results. In general, if the authors are going to use all three measures of size, they should conduct and present parallel analyses on them.
3. I suggest restructuring table 3 so that the linear model corresponding to each row is specified (instead of using the + / - coding that is currently employed. I also suggest that the intercept and parameter estimates be put closer to one another. Having the effect size for the mass ratio surrounded by the codes for whether a variable is present or absent in the model is slightly confusing.
4. 3. Line 320-322: the authors suggest that N. pustulatus is dominant to N. orbicollis when the two species have similar body sizes. I am not sure that I see this in the figure. Can you specify what range of ratios you consider to be similar body sizes?
5. Lines 36-362: It is not clear what is meant here. Can you clarify a bit?

---

## Round 0.2 · Minor Revisions

A reviewer has some additional suggestions to make your presentation clearer. And also, importantly, the reviewer suggests adding a citation supporting the claim that body mass is the "strongest proxy" of body size in burying beetles as well as adding a citation or unpublished data supporting the claim that beetles in the wild arrive at a carcass within minutes of each other (as per your original reply to Reviewer 2).

Reviewer 1 ·

Basic reporting

This revision has addressed many of my comments. Some additional comments below.

L23: Splitting two traits (body size and weapons) with the priority effects in this sentence form seems to suggest that priority effects are also traits. A better structure may be to swap the clause referring to priority effects with the weapons clause.
L35: If body size was the only predictor, then it should state “the largest beetles won”.
L186: It would be more accurate to say just species identity here, as no other traits were measured.
L285: This statement needs a citation. Mass fluctuates with water/food consumption and is therefore not constant for the same individual over time. Mass is fine to use as a proxy, but the statement that it is “the strongest proxy” seems excessive.

Experimental design

L344: Thank you for the explanation about priority effects. I do think an experimental manipulation of priority would have better tested the ideas put forward in the manuscript. I think this methodology would be strengthened yet again if the authors presented some published data concerning the time between visitors to carcasses in nature, as they alluded in their response to reviewers (specifically query 4 from reviewer 2): “…with beetles arriving within minutes of each other…”.

Validity of the findings

All data are code are presented and well annotated.

Additional comments

No comments.

---

## Author Rebuttal · Round 0.2

Dear Dr. Kelly,

Thank you very much for considering our revision of "Behavioral dominance interactions between two species of burying beetle (*Nicrophorus orbicollis* and *Nicrophorus pustulatus*)" (#2022:12:80572:0:1:REVIEW).

Below, we detail responses and changes to the manuscript (in blue) following your comments and comments from the two reviewers. Line numbers refer to the revised (clean) manuscript (not the tracked changes version). We feel that the revised manuscript is much stronger with these changes — we hope that you will find it suitable for publication.

Sincerely,

Yohanna Vangenne, Brendan Sheppard and Paul Martin

Editor comments (Clint Kelly)

MAJOR REVISIONS

Both reviewers were positive about several aspects of your manuscript while also raising some important points that need to be addressed.

First, Reviewer 1 raises an important issue - this manuscript is very similar in many respects to Schrempf et al. 2021 PeerJ 9:e10797. I would think a larger publication studying the interactions of all species would be a stronger paper than publishing what seem to be least-publishable units.

Both Schrempf et al. and this manuscript are undergraduate honours thesis projects, where the lead authors are undergraduates. We submit these projects for publication, rather than waiting to publish them in an amalgamate form, because delaying their publication by several years postpones a key achievement for these students – publishing their work – at a pivotal time in their careers, and amalgamating separate projects dilutes the voices of the lead authors. We note that the current manuscript represents a substantial dataset and effort that includes the detailed transcription of 221 hours of behavioural video from 17 separate trials. Further, the conclusions are well-supported by the data, and the results represent the first-ever documentation of interactions between the two focal species, one of which is ecologically unique among burying beetles, and thus of particular interest.

Second, both reviewers question the measurement and use of "priority" in these current study given that members were simultaneously introduced in each trial. Your methodology and interpretation of the observed effect of this variable need to be justified.

We now clarify and justify our use of 'priority' in the manuscript, and also include a discussion of other potential priority effects and interpretations that we did not address in our work, but could be important in nature.

Third, Reviewer 1 requires some discussion of the potential biases in using wild-caught individuals in the study.

We now include a discussion of this potential bias.

Finally, Reviewer 2 raises a number of statistical issues that need to be addressed.

We have addressed these statistical issues in the revision.

I look forward to reading a revised version of your manuscript.

Thank you!

Reviewer 1 (Anonymous)

Basic reporting

The manuscript was well written. The authors used clear hypotheses and a good structure. I have some concerns about the manuscript, however. First, it seems like a carbon copy of a paper from the same group published in this same journal in 2021 (Schrempf et al. 2021 PeerJ 9:e10797), but using a different burying beetle species. The introduction and literature cited seem identical. On this point alone it is not worthy of rejection, however. The science is sound, but one wonders if a paper will result from replicating this study with every species pair that coexist at the authors' field site.

As we detailed above in response to the general comment from Dr. Kelly, the reason that we have published these two papers independently is that they represent independent undergraduate honours thesis projects that occurred years apart. When graduate students conduct similar experiments, we publish these in combination with other experiments as part of broader papers (e.g., Wettlaufer et al. 2023, Ecological Entomology, which includes behavioural, thermal tolerance, and field experiments on a different beetle species pair).

Second, in the introduction the authors discuss weaponry as a species-level trait that could influence the outcomes of behavioural interactions. While a good example, I think the authors laboured this point—burying beetles do not possess weaponry that is fundamentally different between species. Mandibles are used in fights, but different species do not have drastically different mandibles.

We have removed most of this paragraph when combining the first three paragraphs of the Introduction into one (following Reviewer 1's request below).

Third, priority is not a trait but is referenced as one throughout the manuscript.

We have removed reference to priority as a trait (which we agree, it is not).

In addition, there is nothing in the methods or results about these priority effects. How was it measured or manipulated? I expected the experimental design to include some period of time where an individual was left alone on a carcass to mimic arriving first, with the competitor (smaller or larger) introduced later. Were both individuals introduced at the same time? Any information about priority needs to be rewritten and framed appropriately.

We now include a section in the Methods that details priority effects and how they were measured (lines 218-230; and pasted below).

"Priority effects
We defined 'priority' as the first individual to acquire the resource (the mouse carcass), following previous resource-centered definitions of priority effects (e.g., Poulos & McCormick, 2014), which differ from community assembly definitions (e.g., which species colonizes a new habitat or environment; Fukami, 2015). Beetles were released into containers at the same time, and typically walked around the outside of the containers before burying into the soil. We recorded which individual was first to make contact with the carcass as our estimate of priority, testing the idea that the first beetle to claim the carcass would be better able to retain possession of the carcass over time. Our definition of priority in our trials does not encompass all ways that priority effects could impact burying beetles in nature. For example, beetles that come into contact with the carcass first could benefit by partially burying the carcass, or quickly attracting a mate and defending the carcass as a pair; our trials did not test for such benefits that could also be important in nature."

Experimental design

As noted above, the design of the experiment is not clear regarding the priority effect aspect of the manuscript. If the beetles were introduced at the same time, and priority measured by the first to the carcass, I would not interpret this data as any priority effect.

Our definition of priority is centered on arrival time at the resource (the carcass), but doesn't exclude other potentially important manifestations of priority (e.g., several hours of preparing the carcass before the arrival of a second beetle) that are likely to be important in nature.

All the beetles used in this manuscript were wild caught individuals. The authors did not have any knowledge of whether the individuals they caught were mated or not, or their previous experience with fighting for resources. Could the authors speculate about how these factors,

while unknown, could contribute to the variation in the results they found? With a relatively small sample size, it is possible by chance that their smaller individuals were less likely to have mated which may have been driving the pattern over and above the effect of body size. Some acknowledgement of these issues would be welcome. Why were lab-reared individuals not used?

We used wild-caught beetles, rather than captive-bred beetles, because we did not have access to captive breed individuals of both species, and captive breeding can lead to altered behaviors and other traits, including aggression, compared with wild individuals, which could have altered our results. We now point this out on lines 115-119 (pasted below).

"We used wild-caught beetles, rather than captive-bred beetles, because we did not have access to captive breed individuals of both species, and captive breeding can lead to altered behaviors and other traits, including aggression, compared with wild individuals (Price, 1999; Kelley, Magurran & García, 2006), which could have altered our results."

We did not know the histories of individual beetles used in our experiments, and different histories could have influenced the outcomes of our trials. We now point this out in our methods, and then revisit this possibility in the discussion.

Methods, lines 128-133: "We did not know the individual histories of the beetles used in our trials. For example, some of our beetles could have mated previously or had previous experience in fighting for carcasses. If previous experience varied with the traits of interest (e.g., larger beetles were more likely to have experience fighting), then our results could be biased, leading to incorrect conclusions (e.g., size impacts the likelihood of experience, that then improves fighting success, rather than a direct benefit of large size in fighting). We revisit some of these limitations in our discussion."

Discussion, lines 335-340: "We did not know the histories of wild-caught beetles in our trials (e.g., history of fighting, previous mating); if mass covaried with the history of experience for beetles (e.g., larger beetles had more extensive experience with fighting), then it remains possible that this bias impacted our results. In such a case, the impact of mass would be indirect, manifested through different histories of experience (e.g., fighting experience) or condition (mated or not)."

L150: How did carcass size influence the rate of interactions? Are larger carcasses more fiercely fought over?

Larger carcasses were not associated with more interactions among the beetles (all interactions, $F$ = 0.004, $P$ = 0.95; aggressive interactions, $F$ = 0.057, $P$ = 0.82). We have now added this analysis to the Methods (lines 282-286) and Results (329-330) sections of the manuscript.

L159: Did the individuals have any time in the lab to acclimate?

Yes, we've now added more details on the amount of time beetles were held in captivity to the Methods (lines 133-138):

"The time between a beetle's capture and their participation in a trial averaged 2.8 days (range 2-5 days); beetles in the same trial were usually caught on the same day and held in captivity for the same number of days before trials. We held beetles for at least two days in captivity before trials because we wanted beetles to readjust their behaviors after having been held within the live traps."

and lines 148-150:
"Beetles were selected for trials based on their sex and collection date – either they were collected on the same day (and thus held for the same amount of time; N = 16 trials), or, when this was not possible (N = 1 trial), they were collected within one day of each other."

L170: I do not understand the mass measurement; why is it multiplied by 10g?

Mass was calculated as: weight of the beetle (g) x (10g / weight of the 10g standard (g)) (lines 221-222), not simply multiplied by 10g. Thus, if the 10g standard weighed 9.8g, the beetle weight would be adjusted upward by the appropriate factor (10/9.8 = 1.02) to provide an appropriate estimate of its mass.

L174: Why is 1cm the threshold used for an interaction? Is there any data on the distance when behaviours change in these species? Individuals more than 1cm away might already exhibit different behaviours if they are aware of heterospecifics.

We now provide more information to justify our 1 cm threshold in the Methods (lines 179-184, pasted below). The 1 cm distance was based on previous observations (from Schrempf et al., 2021) where we found evidence for changes in behaviors of beetles at 1cm or less, but did not consistently observe obvious changes in behaviors at greater distances (although subtler changes could have occurred).

"We considered beetles to be interacting when they were within 1 cm of each other because previous work (Schrempf et al., 2021), suggested that beetles were aware of each other at this distance. Beetles may have been aware of each other at greater distances than 1 cm, but we found it difficult to determine if this was consistently true in our trials. Our 1 cm cutoff may thus underestimate the number of interactions among individuals, but was nonetheless consistently applied across trials."

Validity of the findings

This manuscript replicates a previous paper from the same group but has replaced one of the burying beetle species with another. There is little discussion about why the species pair in this manuscript should differ in the mechanism underlying dominance to the previous species pair

of N. tomentosus and N. orbicollis, where similarly body size alone was the significant predictor of dominance.

We have no reason to expect a different mechanism underlying dominance in *N. pustulatus - orbicollis* as compared with *N. tomentosus - orbicollis*. In the Introduction (paragraph 3 of the revised version), we provide evidence that all 3 of the alternative hypotheses may influence who wins carcasses in *Nicrophorus*, and thus any or multiple of these hypotheses could contribute to the outcome of aggressive contests. In addition, *N. pustulatus* is one of the more ecologically-distinct *Nicrophorus* (lines 37-39), which could lead to the dominance interactions among *pustulatus - orbicollis* differing from other *Nicrophorus* species pairs.

The first three paragraphs of the discussion have considerable repetition about body size and dominance. This information should be one paragraph maximum.

We have now combined the first 3 paragraphs of the Introduction into one paragraph.

The discussion is also lacking in broader conclusions. Are the two species tested here very similar in size, or closely related phylogenetically relative to previously published work? What does this paper add to the understanding of mechanisms of dominance?

We have added a discussion of phylogenetic relatedness of our focal species and what our results might mean for the mechanisms of dominance and their persistence over evolutionary time (lines 367-370). We also discuss the relative size of the two species at our study sites and what this size asymmetry might mean for the coexistence of these species within the burying beetle guild (lines 376-379).

All data and code are provided and well annotated.

Additional comments

No comments.

Reviewer 2 (Anonymous)

Basic reporting

This manuscript is clearly written and provides a nice summary of the basic questions it attempts to answer. The authors provide sufficient background to put the manuscript in a broader context. The paper follows a standard structure and the raw data has been provided

(along with a README file). The manuscript I also self contained and for the most part, the conclusions follow from the results.

Thank you!

There is a typo on line 320: "influences" should be "influence".

Corrected - thank you.

Experimental design

The study design is generally sound and the methods are fairly easy to follow. I do have a few comments / suggestions for the authors.
1. I suggest dropping trial 152... it is the only one in which the two beetles were not of the same sex. I imagine that this trial involved a mis-sexed beetle (this happens!). In any case, I suggest removing it form the dataset.

We have now removed this trial from the dataset.

2. Line 157- this suggests that the bicycle light was in the container with the beetles and the carcass. Is this correct?

Yes, the bicycle lights were inside the container with the beetles (preliminary trials with the lights above the containers resulted in too much reflection off the glass that interfered with our video recording). We have now clarified the wording in the Methods (lines 158-160).

3. It is not clear how the pairs were established for their trials. Were the they assigned randomly or were they chosen to generate variation in the body size ratios. I assume the former, but this is not made clear.

Beetle pairs were assigned based on sex and collection date; within these criteria, they were selected haphazardly, without regard to size. We have now added this information to the Methods section (lines 148-150).

4. It sounds as though the beetles were placed in the container at the same time. Doesn't this simulate arriving at the carcass simultaneously (or at least nearly so)? In a more natural situation, the beetles might arrive at the carcass hours apart, thus one species may have begun to prepare the carcass before the other arrives. I wonder whether the authors experiment really simulates that type of priority effects that occur in nature.

Yes, beetles were placed in the container at the same time, and we have added this detail to the Methods (lines 153-154). Once added to the container, beetles typically moved around the outside of the container before burying into the soil. Thus, beetles did not immediately engage with the carcass, but instead engaged with the carcass once they initiated activity themselves in

the evening; our estimates of priority effects address any benefits that a beetle might gain from encountering and engaging with the carcass first (lines 218-229).

In nature, priority effects may take a different form, with beetles arriving at a carcass over greater periods of time; we might have found priority effects if we had allowed beetles to begin to prepare or partially bury the carcass before the second beetle arrived. While our design did not test this kind of priority effect, our field work shows high activity of beetles during key periods when both species are active (e.g. mid June), with extremely high activity just after sunset and all carcasses buried after only two nights. During early evening, beetle activity (mainly *N. orbicollis*) is extremely high, with beetles arriving at some carcasses within minutes of each other, and widespread fighting. Our experiments capture these kinds of interactions, and thus test this mechanism for priority effects, but cannot rule out priority effects manifested in other ways. We now point this out in the manuscript (lines 226-229).
5. This is kind of a minor point, but the authors refer to their study as an "experiment", however nothing is actually manipulated here. Instead, it is an observational study that is conducted under controlled lab conditions.

We have replaced 'experiment' with 'trial' throughout.

Validity of the findings

The analyses and results seem sound. However, I have a few general comments / questions.
1. Lines 248-253: there is a multiple testing issue here (e.g. using three different measures of relative body size, all of which are probably highly correlated, in different models). Have the authors considered adjusting their alpha for this? More generally, I would suggest just picking one measure of size or perhaps constructing a multi-variate measure of size from a PCA.

We included multiple measures of size because reviewers of previous work have asked us to add morphological measures of size, rather than just mass. Our preference is to use mass (only), because selection may favour different morphological traits or proportions in the different species of beetles such that pronotum width and elytra length may not be the best estimates of size between species. With this reasoning, we have now dropped the models with pronotum width and elytra length from the manuscript (although we note that all of the models are significant after correcting for multiple comparisons following Pike 2010, *Methods in Ecology and Evolution* 2:278-282). We also added a sentence describing our reasoning for using mass rather than morphological measures to the methods.

2. 291-299: A related issue- the authors only present the AIC's for the mass model (Table 3). Is there a reason for this? They also note that for all interaction types the results were the same using different measures of body size. They then restrict the analysis to only aggressive interactions. However, when they do this, they only show the mass results. In general, if the authors are going to use all three measures of size, they should conduct and present parallel analyses on them.

We now restrict our analyses to mass as a measure of size.

3. I suggest restructuring table 3 so that the linear model corresponding to each row is specified (instead of using the + / - coding that is currently employed. I also suggest that the intercept and parameter estimates be put closer to one another. Having the effect size for the mass ratio surrounded by the codes for whether a variable is present or absent in the model is slightly confusing.

We have now restructured table 3 following your suggestions.

4. 3. Line 320-322: the authors suggest that N. pustulatus is dominant to N. orbicollis when the two species have similar body sizes. I am not sure that I see this in the figure. Can you specify what range of ratios you consider to be similar body sizes?

A ln ratio of zero indicates identical body sizes, and the model-predicted line sat just below the horizontal 0.5 line (equal wins) when body sizes are identical, suggesting that *N. pustulatus* might win when beetles are the same size. Reanalysis with $N = 17$, however, shows almost no difference, and 95% confidence limits overlap the 0.5 overlap line (equal wins), and so we have removed this suggestion from the Discussion.

5. Lines 36-362: It is not clear what is meant here. Can you clarify a bit?

We decided to delete this sentence because the main point that we were trying to make was already made in the previous sentence.

---

## Round 0.3 · accepted · Accept

Thank you for addressing the additional queries of Reviewer 1. The manuscript reads very well now and I am happy to accept it for publication.